# Early life events in functional abdominal pain disorders in children

**Amaranath Karunanayake[1]***, **Niranga Manjuri Devanarayana[2]**, **Shaman Rajindrajith[3]**

**1** Department of Physiology, University of Ruhuna, Galle, Sri Lanka, **2** Department of Physiology, University of Kelaniya, Ragama, Sri Lanka, **3** Department of Pediatrics, University of Colombo, Colombo, Sri Lanka

* a2222nath@gmail.com

**Data Availability Statement:** All relevant data are within the manuscript and its Supporting Information files.

**Funding:** The author(s) received no specific funding for this work.

## Abstract

### Objectives

Functional abdominal pain disorders (FAPDs) are common gastrointestinal problems in children, and the pathophysiology is thought to be multifactorial. Adverse early life events (ELE) induce alterations in the central nervous system, perhaps predisposing individuals to develop FAPDs. We aimed to study the potential adverse ELE that are associated with FAPDs.

### Methods

We steered a school-based survey involving 1000 children from 4 randomly selected schools. FAPDs were assessed using the translated Rome III questionnaire, and ELE were identified using a pre-tested, parental questionnaire. FAPDs were diagnosed using the Rome III criteria.

### Results

Hundred and eighty-two (182) children had FAPDs (62.1% girls, mean age 8.5, SD 2.1). ELE of them were compared with 571 children without FAPDs (51.1% girls, mean age 8.8, SD 1.9). According to the binary logistic regression analysis, family members with abdominal pain, family member with chronic pain other than abdominal pain, prenatal maternal complications and interventional deliveries, were recognized as potential risk factors for the development of FAPDs. Breast feeding over two years has shown to reduce the prevalence of FAPDs.a.

### Conclusions

Prenatal maternal medical problems are associated a with higher prevalence of FAPDs later in life. Prolonged breastfeeding and normal vaginal delivery could be considered as factors that reduce the vulnerability of developing FAPDs in children. Therefore, minimizing pregnancy-related complications, encouraging vaginal deliveries, and encouraging breastfeeding are potentially valuable measures to prevent FAPDs during childhood.

**Competing interests:** The authors have declared that no competing interests exist.

## Introduction

Functional abdominal pain disorders (FAPDs) is a cluster of ailments consisting of functional dyspepsia (FD), irritable bowel syndrome (IBS), functional abdominal pain non- specified (FAP-NOS), and abdominal migraine (AM). These disorders have a deleterious negative impact on health- related quality of life (HRQoL) of children [1–3], and they are also known to influence the quality of school work adversely [4]. Furthermore, already overstretched pediatric gastroenterology outpatient clinics are overburdened with FAPDs, and the cost of care is also on the rise [5, 6]. Also, poorly managed FAPDs in children continue to have symptoms during adult life [7–10].

Even with cutting-edge science, including molecular medicine and sophisticated imaging techniques, the exact pathophysiological mechanisms of FAPDs are poorly understood. The main pathophysiological mechanisms of FAPDs are thought to be heightened visceral sensitivity, dysmotility of the gastrointestinal tract, mucosal inflammation, altered gut permeability, immunological dysfunction, altered intestinal microbiota, and psycho-social factors [11–13].

Anand *et al*. have demonstrated an association between gastric suction in the neonatal unit and the development of ill-defined gastrointestinal symptoms without overt pathology later in life. They proposed that gastric suction made children vulnerable to developing visceral hyperalgesia, and hypervigilance [14]. Another study using neonatal rats exposed to gastric suction has shown increased visceral and somatic sensitivity, mediated through corticotropin-releasing factor [15]. Several researchers have noted alteration of pain processing pathways when newborns are exposed to painful procedures [16, 17]. These alterations include hypersensitization, reduction in the pain threshold, and permanent alterations in the nociception pathways [18, 19]. Altered pain procession is one of the main pathophysiological mechanisms in FAPDs such as irritable bowel syndrome [20]. In this context, we believed that being exposed to adverse prenatal and post-natal complications would predispose children to develop FAPDs.

Pharmacological interventions for FAPDs are limited and are still in their infancy. Studies have shown that psychological interventions are of value in treating children with FAPDs. Guided imagery and hypnotherapy are effective in treating children with FAPDs [21, 22]. These interventions are promising but challenging to apply in a clinical setting, especially in resource-poor clinics overburdened with patients and lack psychotherapists. Therefore, understanding of ELE that predispose children to FAPDs would give us a window of opportunity to reduce incidence of similar diseases. This paper explores the potential early life events and family-related events that predispose children to develop FAPDs.

## Methods

### Selection of study participants

A cross-sectional study was conducted during 2015 in the schools of the Gampaha District of Sri Lanka. Gampaha district is situated in the Western province of Sri Lanka, and it is the second most populated province of the country. It has both rural and urban areas. The province has 506 schools with both boys and girls. Four such schools (that have classes from grades 1–7) were randomly chosen from the list of schools available in the office of the zonal education director. Two classes from each grade (1–7 years) were selected randomly from those schools. Both random selections were conducted using a random number generator (www.random. org). All children in the selected classes were invited to participate in the study. The average number of students in the class ranged from 20–30.

## Sample size calculation

Previous studies in Sri Lanka have shown that the prevalence of FAPDs is 12.8% [23]. The sample size calculation was based on this prevalence, a confidence level of 95%, power of 80%, and precision found to 5% (d = 0.05). On this basis, the minimum number of children needed was 554. To make the study data more robust, we decided to include 1000 participants.

## Groundwork

The research team visited all four schools before the data collection. The questionnaires were discussed with the principal and the relevant teachers for their approval. The office of the zonal educational director of the district granted permission to conduct the study in these selected schools.

## Data collection

We sent the consent forms, a leaflet that contains details about the study, and questionnaires to parents in a sealed envelope through children in the selected classes. Parents were invited to fill all the questionnaires and the consent forms, seal them in the envelope provided, and return them to school. Contact details of the first author (AK) were provided with the questionnaires, and parents were encouraged to contact the investigator in case clarification was needed. The returned questionnaires were collected through the class teacher. We sent two reminders to parents who did not return the completed questionnaires.

Two questionnaires were used as data collection tools.

1. *Questionnaire on childhood functional gastrointestinal diseases*
   Translated and validated Rome III diagnostic questionnaire for pediatric functional gastrointestinal diseases [24] was used to collect details of gastrointestinal symptoms. This version has been used in several studies [1, 23, 25].

2. *Questionnaire on demographic details and early life events*
   The authors developed this questionnaire and pre-tested in parents who came to hospital clinics with their children (7–12 years old) for minor illness and abdominal pain (**S1 Questionnaires**).

Both questionnaires were in local language (Sinhalese) and the phrases and questions were easy to understand.

## Definitions

FAPDs were diagnosed by using the Rome III criteria for functional gastrointestinal diseases in children/adolescents [26].

## Ethical approval

Ethical approval was obtained from the Ethics Review Committee of the Faculty of Medicine, University of Kelaniya, Sri Lanka.

## Data analysis

We used PSPP version 0.83-g5f9212 statistical software (Free Software Foundation, Inc.htttp//fsf.org/) to analyze data. Frequencies and percentages were calculated for all categorical variables. The independent-sample t-test was used to assess differences between groups. For dichotomous data, the Chi-square test was used to assess differences between the two groups.

Ealry life events that were found to show an association with FAPDs with a p value <0.05 in the univariate analysis were included in the multivariate logistic regression analysis model to calculate the adjusted Odds ratio (aOR) to clearly identify independent factors associated with FAPDs.

## Results

### Sample characteristics

One thousand questionnaires were distributed, and 753 (75.3%) questionnaires with complete data were eligible for the analysis. One hundred eighty-two children fulfilled the criteria for the FAPDs in the 5-12-year age group (mean age 8.53 with a SD of 2.1years). There were 113 (62.1%) girls [mean age of 8.79 years (SD 2.1 years)] in the FAPDs group. The control group consisted of 571 children. There were 292 females (51.1%) [mean age of 8.85 years (SD 2.0 years)] in the control group.

There were 86 children with FAP (47.3%), 48 with IBS (26.4%), 31 with FD (17.0%) and 11 with AM (6%). Six (3.2%) children had more than one FAPDs.

### Parental factors

Having fathers' education level above the ordinary level (OL) (odds ratio (OR) 5.9, 95% confidence interval (CI): 3.44–10.17; $p<$ 0.0001) and mother's education level above the OL (OR 8.15, 95% CI: 4.48–15.15; $p<0.0001$) were significantly associated with FAPDs. Having a father who has an administrative, managerial, and professional job was significantly associated with FAPDs (OR 8.15, 95% CI: 4.48–15.15; $p<0.0001$).

### Family dynamics

The mean number of household members (4.78 [SD = 1.1] vs. 4.80 [SD = 1.2] $p = 0.93$) were not associated with FAPDs. However, family members with chronic abdominal pain were significantly higher among the FAPDs group (23.1% vs. 6.1%) (OR 4.74, 95% CI: 2.91–7.71; $p<0.0001$). Family members with chronic pain other than abdominal pain were also significantly higher in the FAPDs group (29.2% vs. 9.7%) (OR 3.83, 95% CI: 2.04–7.01; $p<0.0001$).

### Prenatal complications

Overall, having prenatal complications predispose children to develop FAPDs (OR 3.95, 95% CI: 2.22–6.64; $p<0.0001$) (Table 1). Gestational diabetes (4.9%) and pregnancy-induced hypertension (4.9%) were the most common ante-natal complications. Period of gestation (36.6 months [SD = 4.5months] in FAPDs vs. 37.5 months [SD = 5.2 months] in controls, $p = 0.120$) was not significantly associated with FAPDs. Interventional deliveries were a risk factor for development of FAPDs (34.1% vs. 18.2%) (OR 2.58, 95% CI: 1.73–3.82; $p<0.0001$).

### Post-natal complication

In univariate analysis overall, postnatal complications were significantly associated with FAPDs (7.7% vs 3.1%) (OR 2.9, 95% CI: 1.77–3.27; $p = 0.036$). Postnatal complications that were significantly associated with FAPDs included neonatal jaundice (OR 2.23, 95%CI 0.83–5.75; $p = 0.049$), and neonatal sepsis (OR 6.39 95%CI 0.9–13.4; $p = 0.0003$).

Table 1. Comparison of ELE between FAPDs and healthy children.

| Risk Factors | FAPDs (n = 182) | Control (n = 571) | P value* |
|---|---|---|---|
| | Mean (SD) | Mean (SD) | |
| Child Age (years) | 8.53 (2.1) | 8.8 (1.9) | 0.112* |
| Gender (Females %) | 62.1 | 51.1 | 0.01# |
| Child's birth order | 1.68 (0.8) | 1.89 (0.9) | 0.011* |
| Mother's age (years) | 36.5 (5.4) | 36.7 (5.7) | 0.686* |
| Mother's Birth order | 2.6 (1.8) | 2.7 (1.9) | 0.387* |
| Maternal age of marriage (years) | 23.6 (4.6) | 23.2 (4.1) | 0.247* |
| Fathers' age (years) | 39.9 (6.8) | 40.1 (6.3) | 0.810* |
| Father's birth order | 2.7 (1.9) | 2.9 (2.4) | 0.179* |
| Paternal age of marriage (years) | 26.9 (4.6) | 26.7 (4.6) | 0.133* |
| Prenatal complications (%) | 14.8 | 7.4 | 0.002# |
| Gestational Diabetes Mellitus | 4.9 | 2.4 | 0.542 |
| Pregnancy Induced hypertension | 4.9 | 2.1 | 0.347 |
| Threatened miscarriages | 1.6 | 1.2 | 0.749 |
| Intra uterine growth retardation | 1.6 | 1.0 | 0.472 |
| Other diseases | 1.6 | 0.7 | 0.483 |
| Post-natal complications (%) | 7.7 | 3.1 | 0.008# |
| Neonatal Jaundice | 5.5 | 2.1 | 0.0491 |
| Sepsis | 2.2 | 0.18 | 0.0003 |
| Interventional/assisted delivery | 34.1 (2.1) | 18.2 (.2.3) | 0.000# |
| SCBU care (%) | 7.1 | 2.9 | 0.004# |
| Post-delivery hospital stay (days) | 3.3 (0.9) | 2.5 (1.0) | 0.933* |
| 4 months exclusive breast feeding (%) | 66.6 | 61.3 | 0.301# |
| Breast feeding more than 12 months (%) | 51.2 | 50.3 | 0.51# |
| Breast feeding more than 24 months (%) | 23.6 | 50.3 | 0.000# |

*Independent-sample T test

#Chi square, SCBU—Special Care Baby Units, FAPDs- Functional Abdominal Pain Disorders.

### Breastfeeding

Exclusive breast feeding for more than 4 months (66.6% vs. 61.3%) or breast feeding more than 1 year (51.2% vs 50.3%) was not significantly different between the groups. However, breast feeding more than 2 years was a significantly lower in the FAPDs group (23.6 vs 50.3).

After binary logistic regression analysis and calculation of adjusted Odds ratio (aOR), family members with abdominal pain, family member with chronic pain other than abdominal pain, prenatal complications and, interventional deliveries were recognized as potential risk factors for the development of FAPDs. Breast feeding over two years has shown to reduce the prevalence of FAPDs (Table 2).

### Discussion

Exposure to early adverse life events makes children and adults more vulnerable to develop functional gastrointestinal disorders. However, they have not been systematically studied in relation to children. This paper reports several early prenatal factors that could potentially predispose children to develop FAPDs. Besides, we noted that parental factors, such as higher social class and family history of abdominal pain, were also associated with FAPDs. Prolonged breastfeeding over two years appears to be associated with lower prevalence of FAPDs.

**Table 2. Multivariate analysis to identify risk and protective factors for development of FAPDs.**

| Factor related to development of the disease | Disease status | | | | |
|---|---|---|---|---|---|
| | FAPDs (n = 182) | Healthy (n = 571) | Total Number (%) | aOR (95% CI) | Significance |
| | Number (%) | Number (%) | | | |
| No family Member with abdominal pain (R) | 139 (76.4%) | 536(93.9%) | 675(89.6%) | 4.756(2.785–8.121) | P<0.0001 |
| Family Member with abdominal pain | 43(23.6%) | 35(6.1%) | 78(10.4%) | | |
| No family Member with chronic pain other than abdominal pain (R) | 129 (70.9%) | 516 (90.4%) | 645(85.7%) | 2.805(1.777–4.427) | P<0.0001 |
| Family Member with chronic pain other than abdominal pain | 53 (29.1%) | 55(9.6%) | 108(14.3%) | | |
| No prenatal Complications (R) | 155(85.2%) | 529 (92.6%) | 684 (90.8%) | 1.940(1.086–3.468) | P = 0.025 |
| Prenatal Complications | 27(14.8%) | 42(7.4%) | 69 (9.2%) | | |
| Normal vaginal delivery | 120 (65.9%) | 467 (81.8%) | 587 (77.9%) | 2.184 (1.450–3.288) | P<0.0001 |
| Deliveries other than vaginal delivery | 62 (34.1%) | 104 (18.2) | 166 (22%) | | |
| No SCBU care (R) | 169(92.9%) | 555 (97.2%) | 724 (96.1%) | 1.279(0.091–17.955) | P = 0.855 |
| SCBU care | 13(7.1%) | 16(2.8%) | 29 (3.9%) | | |
| No Postnatal complications (R) | 168(92.3%) | 553 (96.8%) | 721 (95.8) | 2.123 (0.171–26.304) | P = 0.558 |
| Postnatal complications | 14 (7.7%) | 18 (3.2%) | 32(4.2%) | | |
| Breast feeding more than 24 months (R) | 43 (23.6%) | 288 (50.4%) | 331(44.0%) | 0.275(0.182–0.416) | P <0.0001 |
| No breast feeding more than 24 months | 139 (76.4%) | 283 (49.6%) | 422(56.0%) | | |

R- Reference category, aOR- adjusted Odds ratio, CI- 95% confidence interval, FAPDs- Functional Abdominal Pain Disorders, SCBU—Special Care Baby Units.

Adverse life events in early life have been recognized as potential predisposing factors for IBS in adults. It had been shown that adverse social factors, exposure to emotional, physical, and sexual abuse below 18 years were linked with IBS in adults [27]. Furthermore, VIdelock et al. also found that adults who suffered adverse life events during childhood had a higher cortisol response after a stressful event [28]. Further to this, Anand et al., reported exposure to gastric suction in the early newborn period increase the risk of developing symptoms suggestive of FAPDs later in life [14]. In their study, Anand et al. did not report a link between admission to special care baby units (SCBUs) and FAPDs. Similar to this, we also did not observe a significant association between admission to SCBUs and the development of FAPDs.

In this study, we noted that overall prenatal complications were significantly associated with FAPDs later in childhood. However, individual factors such as pregnancy-induced hypertension, gestational diabetes, and intrauterine growth retardation were not associated with the development of FAPDs, possibly due to the small numbers in each category. None of the previous studies has shown a link between prenatal complications and FAPDs. In addition to these factors, we also identified delivery mode other than normal vaginal delivery as a potential risk factor for developing FAPDs. It is possible that newborns with prenatal complications are ended up in SCBUs and being subjected to invasive procedures which could induce pain.

Exposure to pain during these procedures in the neonatal period leads to irreversible alterations in the pain processing pathways altering pain perception even up to adulthood [16, 17, 29]. In addition, exposure to tissue-damaging interventions, repeated regularly over days to weeks, adversely affect the development of nociceptive circuitry in the central level and contributed to alteration in pain processing pathways that could last throughout the life [16, 30–33].

Exposure to noxious stimuli during the neonatal period leads to sensitization of pain processing pathways, which could permanently alter nociception throughout life [32]. It had also been proposed that exposure to stressful and painful experiences in the SCBU could also alter the hypothalamo-pituitary-adrenal axis leading to excess release of adrenaline and cortisol [26]. Therefore, we hypothesized admission to SCBU and postnatal complications

predisposing graduates from SCBUs to develop FAPDs later in life through visceral hypersensitivity, hypervigilance for pain, and altered pain processing at the central level. In addition, receiving antibiotics in SCBUs alter the microbiome ard the brain- gut-mircrobiome axis of these neonates leading them to develop FAPDs [19, 34]. Most neonates admitted to SCBUs receive antibiotics therapeutically or prophylactically. Receiving intravenous antibiotics in early life is likely to interfere with the development of the intestinal microbiota of the newborn. It is well known that early antibiotic therapy leads to reduced colonization of Bifidobacteria, Bacteroids and increases Fermicutis species leading to significant derangement of bacterial diversity [34]. Altered microbiota is a well-known risk factor for the development of FAPDs in children. Children with IBS are known to have increased colonies of Enterobacteriaceae, Verionella and Dorea species and lower number of colonies of Bifidobacterium, Collinsella and Clostridiales [19]. *Haemophilus parainfluenzae* were noted to be represented in higher numbers in children with IBS [35, 36]. Therefore, the alteration of intestinal microbiota may contribute to the development of FAPDs in children. Separation of the newborn from the mother by admitting them to an SCBU is another possible reason for them to have a higher predilection to develop FAPDs. In animal studies, it was proven that the separation of newborn puppies from their mother could lead to the development of features of IBS in them [37]. However, we did not find association between admission to SCBUs or postnatal complications and FAPDs. Small numbers of children who had experienced these events would have been the reason for this lack of association.

In this study, non-vaginal delivery was identified as a significant risk factor in developing FAPDs. Lower segment caesarian sections are known to alter the intestinal microbiota. Vaginal delivery helps the newborn's intestine get colonized with vaginal microflora dominated by Lactobacilli and Bacteroids [38]. It is reported that babies born through a caesarian section have reduced Bifidobacteria and Bacteroids and increased numbers of Propionibacteria, Staphylococci, and Corynebacteria, which incidentally dominate the microflora of the skin [34]. As mentioned earlier, altered microbiota may contribute to the development of FAPDs through several mechanisms other than the subtle alteration of the microbiome alone. They include increased intestinal permeability, altered brain function, modulation of the enteric nervous system, changes in motility patterns of the gastrointestinal tract, and heightened visceral sensitivity [39].

Exclusive breastfeeding is recommended for all babies up to 6 months, and non-exclusive breastfeeding should be continued for as long as mutually desired by mother and child [40]. The advantages of breastfeeding in preventing diseases had been highlighted in many reviews [41]. Suboptimal breastfeeding is associated with large sums of direct and indirect medical costs [42]. In this study, we found a lower prevalence of FAPDs in children who had prolonged breastfeeding over two years. Although breastfeeding is known to prevent many gastrointestinal diseases, including gastrointestinal infections, necrotizing enterocolitis, coeliac disease, and inflammatory bowel disease, no study has reported its ability to reduce the prevalence of FAPDs. It is possible to explain this phenomenon in several ways. It is well known that FAPDs are associated with gastrointestinal allergies, and prolonged breastfeeding can prevent gastrointestinal allergies [40]. In addition, it had been described that the intestinal microbiome of breastfed children (rich in Bifidobacteria, Lactobacilli) is different from that of children fed otherwise (rich in Staphylococci, Enterobacteriaceae, and Veillonella) [43]. Adult-like microbiota is known to get established during the second and third years of life. Extended breastfeeding over two years of age could positively influence this process and may have contributed to preventing FAPDs in children.

We also noted that children with a family history of abdominal pain and other chronic pain disorders have a higher predilection to develop FAPDs. Intergenerational transmission of

gastrointestinal symptoms is a well-known phenomenon. Children with a mother suffering from IBS are more likely to have troublesome gastrointestinal symptoms, poor school attendance due to abdominal pain. They have a higher need for healthcare consultation for gastrointestinal symptoms [44, 45]. The mechanisms for this phenomenon are not entirely evident. Intergenerational transmission of pain behavior may occur through social learning and modeling of pain in family members. Furthermore, better parental education and higher social status according to paternal profession were also associated with FAPDs. Contrary to this, in a previous study we noted a higher prevalence of FAPDs in lower socioeconomic strata[23]. However, a meta-analysis on epidemiology of FAPDs in children failed to demonstrate a statistically significant association between socioeconomic status and FAPDs [42]. Similarly, a systematic review among adults with IBS covering worldwide data also found no association between IBS and socioeconomic status [46].

We recruited a large number of students from schools to get an adequate sample to provide us statistically valid inferences. All FAPDs were defined using the Rome III criteria for children and adolescents. However, our sample was not large enough to identify the individual risk factors for postnatal complications. Also, since this was an epidemiological survey, we could not perform a complete physical examination of children who participated. Most of the published studies on epidemiological surveys, risk factors for FAPDs, and recurrent abdominal pain have not conducted a physical examination [47]. It can also be argued that there could be a recall bias in this study, as most of the events that were evaluated happened several years ago. However, the data we requested were simple, and we believe that they are major life events of a family that most parents would recall without much of a problem. Finally, there could be other confounding factors such as psychological stress, exposure to surgical interventions, and gastrointestinal infections we have not included in the study that could have contributed to the development of FAPDs.

We conclude by stating that FAPDs are a common problem in young children. Prenatal maternal medical problems are associated with higher prevalence of FAPDs later in life. Prolonged breastfeeding and normal vaginal delivery could be considered as factors that reduce the vulnerability of developing FAPDs in children. Therefore, minimizing pregnancy-related complications, encouraging vaginal deliveries, and encouraging breastfeeding are potentially valuable measures to prevent FAPDs during childhood.

## Supporting information

**S1 Questionnaires.**
(PDF)

**S1 Dataset.**
(SAV)

## Acknowledgments

An Abstract on this article was published as Impact of early life events (ELE) on sex-related vulnerability in developments of functional abdominal pain disorders (FAPDs) in 5–12 age group Amaranath Karunanayake, Niranga Manjuri Devanarayana, Shaman Rajindrajith: The 14th Asian Pan-Pacific Society of Paediatric Gastroenterology, Hepatology and Nutrician Meeting, Bangkok, Thailand (2018) Abstract no. PP-G-16.

The conference proceedings were peer-reviewed and published as a poster /abstract presentation.

## Author Contributions

**Conceptualization:** Amaranath Karunanayake, Niranga Manjuri Devanarayana, Shaman Rajindrajith.

**Data curation:** Amaranath Karunanayake.

**Formal analysis:** Amaranath Karunanayake.

**Investigation:** Amaranath Karunanayake.

**Methodology:** Amaranath Karunanayake, Niranga Manjuri Devanarayana, Shaman Rajindrajith.

**Supervision:** Niranga Manjuri Devanarayana.

**Writing – original draft:** Amaranath Karunanayake.

**Writing – review & editing:** Niranga Manjuri Devanarayana, Shaman Rajindrajith.

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
