## [Decision Letter · Decision Letter 0]

13 Sep 2020

PONE-D-20-15012

Early life events in functional abdominal pain disorders in children

PLOS ONE

Dear Dr. Karunanayake,

Thank you for submitting your manuscript to PLOS ONE. After careful consideration, we feel that it has merit but does not fully meet PLOS ONE’s publication criteria as it currently stands. Therefore, we invite you to submit a revised version of the manuscript that addresses the points raised during the review process.

We look forward to receiving your revised manuscript.

Kind regards,

Zhen Hua Hu, MD, PhD

Academic Editor

PLOS ONE

Journal Requirements:

2. Please change your reference to "p=0.000" to "p<0.001" or as similarly appropriate, as p values cannot equal zero.

3. Please include your tables as part of your main manuscript and remove the individual files. Please note that supplementary tables (should remain/ be uploaded) as separate "supporting information" files.

4. We noted in your submission details that a portion of your manuscript may have been presented or published elsewhere.

"An Abstract on this article was published as  

Impact of early life events (ELE) on sex-related vulnerability in developments of functional abdominal pain disorders (FAPDs) in 5-12 age group Amaranath Karunanayake, Niranga Manjuri Devanarayana, Shaman Rajindrajith: The 14th   Asian Pan-Pacific Society of Paediatric Gastroenterology, Hepatology and Nutrician Meeting, Bangkok, Thailand (2018) Abstract no. PP-G-16."

Reviewers' comments:

Reviewer's Responses to Questions

**Comments to the Author**

1. Is the manuscript technically sound, and do the data support the conclusions?

Reviewer #1: Partly

Reviewer #2: Yes

2. Has the statistical analysis been performed appropriately and rigorously? 

Reviewer #1: Yes

Reviewer #2: Yes

3. Have the authors made all data underlying the findings in their manuscript fully available?

Reviewer #1: No

Reviewer #2: Yes

4. Is the manuscript presented in an intelligible fashion and written in standard English?

Reviewer #1: No

Reviewer #2: Yes

5. Review Comments to the Author

Reviewer #1: First of all, congratulations to the authors for the interesting paper. The findings of the study are really remarkable and the outcomes would be of much value in the pediatric patients for the prevention of Functional Abdominal Pain Disorders.

There are few comments from my side on the paper which I think should be made clear for the further validity and reliability of the paper before publishing.

1. My first comment is on the criteria used for the diagnosis of FAPD. There has been recent update on the ROME III criteria to ROME IV criteria with some significant changes. Is there any rationale for the use of ROME III criteria and not ROME IV criteria in the study?

2. My second comment is on the methodology regarding sampling methods followed in the study. How was the random selection of classes and students done?

Also is the sample size adequate? Is there any basis or rationale for the given sample size in the study?

3. The questionnaire used for data collection is not provided as supplementary files. Could you please provide the validated questionnaire used in the study?

4. Is there any basis for significant relationship of parental factors like education level and jobs related to the FAPD in the children? They are found to be statistically significant in the study but not discussed in discussion section.

5. There are many typographical errors in the manuscript. For example, in results section sample characteristics sub heading the following second sentence is not clear.

"The control group consisted of 571 children. There were 292 (51.1%) [mean age of 8.85 years (SD 2.0 years)] in the control group."

Font of the text is also not uniform in the manuscript and there are grammatical errors.

Reviewer #2: Introduction:

In paragraph 3: authors tried to relate or see commonalities between gastric suction and adverse prenatal and post-natal complications that would predispose children to develop FAPDs…..this need to be explained how?

In paragraph 4: psychological interventions are promising but challenging to apply in a clinical setting…..which psychological interventions are being referred here? If would be nice if this can get properly referred?

Therefore, understanding of ELEs that predispose children to FAPDs would

give us a window of opportunity to prevent them from developing similar diseases……should be replaced by reducing the incidence of similar disease.

Methods:

Four mixed schools that had classes from grade 7 or above were selected randomly from those schools by drawing lots…….not understood fully what “drawing lots” means?

“Two classes from each grade (grade 1-7) were selected randomly from those selected schools”…..just a sentence before it’s being said grade 7 and above? This should be corrected by clearly stating is it grade 7 and above or between grade 1-7.

Data collection:

The following information seems redundant:

“Children handed over the questionnaires to the class teacher who collected them. Investigators collected all returned questionnaires from the class teachers. We sent two reminders to parents who did not return the completed questionnaires”

Results:

“A thousand questionnaires were distributed”……I think it should be replaced by One thousand questionnaires were distributed.

“The control group consisted of 571 children. There were 292 (51.1%) [mean age of 8.85 years (SD 2.0 years)] in the control group” ……..Either there is a mistake here or I am confused??? Is the number for control group 571 or 292 children???

“There were 86 children with FAP (47.3%), 48 with IBS (26.4%), 31 with FD (17.0%) and 11

with AM (6%). Six (3.2 %) children had more than one AP-FGIDs”……should it be FAPDs

Discussion:

On Page 19 line 1 “In addition to these factors, we also identified the mode of delivery other than normal

vaginal delivery was also identified as a probable risk factor for developing” FAPDs……..highlighted “was also identified” should be deleted from this sentence.

On page 20, in 2nd paragraph, “Receiving intravenous antibiotics in early neonatal period is likely to interfere with the development of the gut microbiota and the brain-gut-microbiota axis of the newborn”. ……please replace with “Receiving intravenous antibiotics in early neonatal period is likely to interfere with the development and possible alteration of the gut microbiota and the brain-gut-microbiota axis of the newborn”.

6. PLOS authors have the option to publish the peer review history of their article (what does this mean?). If published, this will include your full peer review and any attached files.

Reviewer #1: **Yes: **Purushottam Adhikari

Reviewer #2: No

---

## [Author Response · Author response to Decision Letter 0]

29 Dec 2020

Reviewer #1: 

Comment

First of all, congratulations to the authors for the interesting paper. The findings of the study are really remarkable and the outcomes would be of much value in the pediatric patients for the prevention of Functional Abdominal Pain Disorders.

Response

Thank you 

There are few comments from my side on the paper which I think should be made clear for the further validity and reliability of the paper before publishing.

Comment 

1. My first comment is on the criteria used for the diagnosis of FAPD. There has been recent update on the ROME III criteria to ROME IV criteria with some significant changes. Is there any rationale for the use of ROME III criteria and not ROME IV criteria in the study?

Response:

We conducted this study in 2015. During that time Rome IV criteria were not available. Therefore, we used Rome III criteria which were the latest at that time. When the newly generated Rome IV criteria were released in 2016, there was a major change in definitions of all abdominal pain related functional gastrointestinal disorders. That was the reduction of duration of symptoms from two months to one month. Therefore, we were unable to use Rome IV criteria

Comment 

2. My second comment is on the methodology regarding sampling methods followed in the study. How was the random selection of classes and students done?Also is the sample size adequate? Is there any basis or rationale for the given sample size in the study?

Response:

Details about sample calculation was included into the manuscript. The necessary sample size was 554. To make our data more robust we included 1000 children. Random selection of schools and classes were done by using a random number generator. (www.random.org)

Sample size and sampling technique

The sample size for this study was calculated by using the following formula (Lowanga, Lameshow 1991).

Size or the sample N = Z2 P (1-P) /d2

 Z = 1.96

 P = anticipated population proportion 

 D = absolute precision required for the estimate to fall within a given percentage point of the proportion = 2.5%. 

 10% was taken as the value for P from previous Sri Lankan study (Devanarayana, Mettananda et al. 2011).

 P=10%

N = Z2 P (1-P) /d2

=1.96*1.96*0.9/0.025*0.025= 3.45744/0.00625 = 553.19

Substituting the above values in the equation the required sample size proved to be 553. The response rate was anticipated as 60%. Therefore, to achieve the desired sample size, it was decided to recruit 1000 school children. 

Comment

3. The questionnaire used for data collection is not provided as supplementary files. Could you please provide the validated questionnaire used in the study?

Response

We included the questionnaires used in this study as supplementary materials

Comment

4. Is there any basis for significant relationship of parental factors like education level and jobs related to the FAPD in the children? They are found to be statistically significant in the study but not discussed in discussion section.

Response

In the revised manuscript, we discussed these factors and their relevance.

Comment

5. There are many typographical errors in the manuscript. For example, in results section sample characteristics sub heading the following second sentence is not clear.

"The control group consisted of 571 children. There were 292 (51.1%) [mean age of 8.85 years (SD 2.0 years)] in the control group."

Font of the text is also not uniform in the manuscript and there are grammatical errors.

Response

We corrected all typographic and grammatical errors in the revised version. 

Reviewer #2:

 Introduction:

Comment:

In paragraph 3: authors tried to relate or see commonalities between gastric suction and adverse prenatal and post-natal complications that would predispose children to develop FAPDs…..this need to be explained how?

Response

Paragraph 3 was changed. We included more detailed explanation regarding neonatal pain related events and the logical reasons for conduct the study.

Comment

In paragraph 4: psychological interventions are promising but challenging to apply in a clinical setting…..which psychological interventions are being referred here? If would be nice if this can get properly referred?

Response

We included details about psychological interventions of childhood FAPDs and appropriate references to the paragraph.

Comment

Therefore, understanding of ELEs that predispose children to FAPDs would

give us a window of opportunity to prevent them from developing similar diseases……should be replaced by reducing the incidence of similar disease.

Response:

Corrected as suggested by the reviewer

Methods:

Comment

Four mixed schools that had classes from grade 7 or above were selected randomly from those schools by drawing lots…….not understood fully what “drawing lots” means?

Response

Details about the random selection of schools and classes were included into the manuscript.

Comment

“Two classes from each grade (grade 1-7) were selected randomly from those selected schools”…..just a sentence before it’s being said grade 7 and above? This should be corrected by clearly stating is it grade 7 and above or between grade 1-7.

Response

The section was re-written with more details and the error was corrected. The grades were grades 1 to 7 not grade 7 or above. We would like to thank the reviewer for pointing out this mistake.

Data collection:

Comment

The following information seems redundant:

“Children handed over the questionnaires to the class teacher who collected them. Investigators collected all returned questionnaires from the class teachers. We sent two reminders to parents who did not return the completed questionnaires”

Response:

The sentences were edited.

Results:

Comment

“A thousand questionnaires were distributed”……I think it should be replaced by One thousand questionnaires were distributed.

Response

Corrected as suggested by the reviewer.

Comment

“The control group consisted of 571 children. There were 292 (51.1%) [mean age of 8.85 years (SD 2.0 years)] in the control group” ……..Either there is a mistake here or I am confused??? Is the number for control group 571 or 292 children???

Response

We thank the reviewer for pointing out this mistake. It was corrected.

It should have been “The control group consisted of 571 children. There were 292 males (51.1%)”

Comment

“There were 86 children with FAP (47.3%), 48 with IBS (26.4%), 31 with FD (17.0%) and 11

with AM (6%). Six (3.2 %) children had more than one AP-FGIDs”……should it be FAPDs

Response 

Corrected as suggested by the reviewer

Discussion:

Comment

On Page 19 line 1 “In addition to these factors, we also identified the mode of delivery other than normal

vaginal delivery was also identified as a probable risk factor for developing” FAPDs……..highlighted “was also identified” should be deleted from this sentence.

Response

Corrected as suggested by the reviewer

Comment

On page 20, in 2nd paragraph, “Receiving intravenous antibiotics in early neonatal period is likely to interfere with the development of the gut microbiota and the brain-gut-microbiota axis of the newborn”. ……please replace with “Receiving intravenous antibiotics in early neonatal period is likely to interfere with the development and possible alteration of the gut microbiota and the brain-gut-microbiota axis of the newborn”.

Response

Corrected as suggested by the reviewer

---

## [Decision Letter · Decision Letter 1]

26 May 2021

PONE-D-20-15012R1

Early life events in functional abdominal pain disorders in children

PLOS ONE

Dear Dr. Karunanayake,

Thank you for submitting your manuscript to PLOS ONE. After careful consideration, we feel that it has merit but does not fully meet PLOS ONE’s publication criteria as it currently stands. Therefore, we invite you to submit a revised version of the manuscript that addresses the points raised during the review process.

We look forward to receiving your revised manuscript.

Kind regards,

Zhen Hua Hu, MD, PhD

Academic Editor

PLOS ONE

Reviewers' comments:

Reviewer's Responses to Questions

**Comments to the Author**

1. If the authors have adequately addressed your comments raised in a previous round of review and you feel that this manuscript is now acceptable for publication, you may indicate that here to bypass the “Comments to the Author” section, enter your conflict of interest statement in the “Confidential to Editor” section, and submit your "Accept" recommendation.

Reviewer #1: (No Response)

Reviewer #3: (No Response)

2. Is the manuscript technically sound, and do the data support the conclusions?

Reviewer #1: Yes

Reviewer #3: Partly

3. Has the statistical analysis been performed appropriately and rigorously? 

Reviewer #1: Yes

Reviewer #3: No

4. Have the authors made all data underlying the findings in their manuscript fully available?

Reviewer #1: Yes

Reviewer #3: No

5. Is the manuscript presented in an intelligible fashion and written in standard English?

Reviewer #1: Yes

Reviewer #3: Yes

6. Review Comments to the Author

Reviewer #1: Thank you for addressing the comments!

I have one more issue and I think that needs to be addressed. After looking the questionnaires I could see many technical terms used there in questionnaire like vertigo, elective cesarean section, emergency cesarean section and so on. Has the questionnaire been delivered in local language or English?

As the questionnaire is self administered, how was this addressed?

If the local language translation were used, so far my knowledge it should be mentioned in the methods section.

Reviewer #3: This is an interesting study on the association between certain early life events and functional abdominal pain disorders (FAPDs) in children. The following are my comments:

• In the methods: Authors should clarify when the study was conducted.

• Methods; Selection of study participants: “Four such schools (that have classes from grades 1-7) were randomly chosen …... Two classes from each grade (1-7 years) were selected randomly…All children in the selected classes were invited to participate in the study”. How did the authors accurately include invited 1000 participants? Average number of students per class?

• Methods; Data collection: “Parents were invited to fill all the questionnaires and the consent forms, seal them in the envelope provided, and return them to school”. Non-response may be attributed to parents being illiterate (selection bias)?

• Methods; Data analysis: Authors have to clarify all independent variables analyzed in their study, including composite variables (e.g., pre-natal complications, post-natal complications) and used cutoffs (e.g., parental education; ordinary level?). Being the main focus in this study, pre-natal and post-natal complications should also be individually reported.

• The authors didn’t report whether there are statistically significant differences between children with and without FABDs in some important characteristics, such as gender and residence.

• Authors investigated the association of several independent variables with FAPDs, and found some statistically significant relations using bivariable analysis. However, multivariable statistical analysis (e.g., logistic regression) is required to control possible confounding factors.

• Moreover, authors should explain in the discussion the possibility of other confounding factors that are not included in this study, such as psychological factors, surgeries, and infections.

• In the discussion: “we noted a significant association between admission to SCBUs and the development of FAPDs”. However, the authors didn’t report this in the results section.

• The study design can allow determining associations but not causal-relationships. That is, authors should revise wordings, such as: In the abstract: “The extended period of breastfeeding (>2years) was shown to reduce the occurrence of FAPDs”.

• Authors should acknowledge, in the manuscript, their use of ROME III criteria, rather than the most updated ROME IV criteria, as one of the study limitations.

7. PLOS authors have the option to publish the peer review history of their article (what does this mean?). If published, this will include your full peer review and any attached files.

Reviewer #1: No

Reviewer #3: **Yes: **Elsayed Abdelkreem

---

## [Author Response · Author response to Decision Letter 1]

30 Jul 2021

Reviewer 1 comments Response 

Reviewer #1: Thank you for addressing the comments! Your comments were very valuable, and we believe they made our manuscript more strong.

I have one more issue and I think that needs to be addressed. After looking the questionnaires I could see many technical terms used there in questionnaire like vertigo, elective cesarean section, emergency cesarean section and so on. Has the questionnaire been delivered in local language or English? The questionnaire was developed in English and translated to the local language (Sinhalese). The questions were simple, and the technical terms were carefully translated so that a simple layman can understand the meaning. We pretested it in the hospital clinic and parents were able to understand the questionnaire without difficulty.

In addition, the telephone number of the first author was in the questionnaire and we encouraged the parents to call should they need clarifications regarding any question or a phrase.

As the questionnaire is self administered, how was this addressed? Please see above.

If the local language translation were used, so far my knowledge it should be mentioned in the methods section. We agree with the reviewer and a sentence was added into the method section stressing the language.

PONE-D-20-15012R1

Early life events in functional abdominal pain disorders in children

PLOS ONE

Second revision 

Responses for reviewers comments for version 2

Reviewer 3 comments Response 

This is an interesting study on the association between certain early life events and functional abdominal pain disorders (FAPDs) in children. The following are my comments: Thank you.

In the methods: Authors should clarify when the study was conducted. The study was conducted in 2015. We included it into the manuscript.

• Methods; Selection of study participants: “Four such schools (that have classes from grades 1-7) were randomly chosen …... Two classes from each grade (1-7 years) were selected randomly…All children in the selected classes were invited to participate in the study”. How did the authors accurately include invited 1000 participants? Average number of students per class? When we calculated sample size, the minimum number of children needed for the study came as 554. To make study data more robust we included 1000 participants. The average number of children in classes ranged from 20-35.

• Methods; Data collection: “Parents were invited to fill all the questionnaires and the consent forms, seal them in the envelope provided, and return them to school”. Non-response may be attributed to parents being illiterate (selection bias)? The literacy rate in Sri Lanka is >95% and it is extremely unlikely that both parents are illiterate in a given home. The questionnaire was in the local language, and we developed it very carefully that an average person could understand it without any difficulty. It was noted that the questionnaire could be understood by an average person without a difficulty during pretesting.

• Methods; Data analysis: Authors have to clarify all independent variables analyzed in their study, including composite variables (e.g., pre-natal complications, post-natal complications) and used cutoffs (e.g., parental education; ordinary level?). Being the main focus in this study, pre-natal and post-natal complications should also be individually reported. In the revised manuscript key pre and postnatal events were individually reported (table 1) 

The authors didn’t report whether there are statistically significant differences between children with and without FABDs in some important characteristics, such as gender and residence. Table 1 depicts all demographic variables and in that table, most of the sociodemographic factors of the cases and controls were compared. The study was conducted in a semi-urban setting and therefore residence was not included in the comparison as there would not be a difference in this variable. 

• Authors investigated the association of several independent variables with FAPDs, and found some statistically significant relations using bivariable analysis. However, multivariable statistical analysis (e.g., logistic regression) is required to control possible confounding factors. In the revised manuscript, Table 2 depicts the binary logistics regression to determine the early life events as the predictors of development of FAPDs

Moreover, authors should explain in the discussion the possibility of other confounding factors that are not included in this study, such as psychological factors, surgeries, and infections. This valuable suggestion was taken into consideration and a sentence was included into the limitations

• In the discussion: “we noted a significant association between admission to SCBUs and the development of FAPDs”. However, the authors didn’t report this in the results section. This variable is in table 1.

• The study design can allow determining associations but not causal-relationships. That is, authors should revise wordings, such as: In the abstract: “The extended period of breastfeeding (>2years) was shown to reduce the occurrence of FAPDs”. We agree with the reviewer. The wording implying casual-relationship was changed to associations

• Authors should acknowledge, in the manuscript, their use of ROME III criteria, rather than the most updated ROME IV criteria, as one of the study limitations. When we conducted the study in 2015 the available version of Rome criteria was Rome III. Therefore, we do not see it as a limitation.

---

## [Decision Letter · Decision Letter 2]

7 Oct 2021

PONE-D-20-15012R2Early life events in functional abdominal pain disorders in childrenPLOS ONE

Dear Dr. Karunanayake,

Thank you for submitting your manuscript to PLOS ONE. After careful consideration, we feel that it has merit but does not fully meet PLOS ONE’s publication criteria as it currently stands. Therefore, we invite you to submit a revised version of the manuscript that addresses the points raised during the review process. While reviewer 1 is happy with the revised manuscript, some of reviewer 3's concerns remain to be addressed. 

We look forward to receiving your revised manuscript.

Kind regards,

Zhen Hua Hu, MD, PhD

Academic Editor

PLOS ONE

Reviewers' comments:

Reviewer's Responses to Questions

**Comments to the Author**

1. If the authors have adequately addressed your comments raised in a previous round of review and you feel that this manuscript is now acceptable for publication, you may indicate that here to bypass the “Comments to the Author” section, enter your conflict of interest statement in the “Confidential to Editor” section, and submit your "Accept" recommendation.

Reviewer #1: All comments have been addressed

Reviewer #3: (No Response)

2. Is the manuscript technically sound, and do the data support the conclusions?

Reviewer #1: Yes

Reviewer #3: No

3. Has the statistical analysis been performed appropriately and rigorously? 

Reviewer #1: Yes

Reviewer #3: No

4. Have the authors made all data underlying the findings in their manuscript fully available?

Reviewer #1: Yes

Reviewer #3: Yes

5. Is the manuscript presented in an intelligible fashion and written in standard English?

Reviewer #1: Yes

Reviewer #3: Yes

6. Review Comments to the Author

Reviewer #1: The comments have been addressed well by the authors. So far on my knowledge, the manuscript has been well organized with the details than earlier version submitted.

Reviewer #3: The authors successfully addressed most reviewers' comments. However, authors should clarify statistical methods used in Table 2, why these variables were specifically included, and what are the interpretation of results?

7. PLOS authors have the option to publish the peer review history of their article (what does this mean?). If published, this will include your full peer review and any attached files.

Reviewer #1: No

Reviewer #3: **Yes: **Elsayed Abdelkreem

---

## [Author Response · Author response to Decision Letter 2]

29 Nov 2021

1. An Abstract on this article was published as 

‘’Impact of early life events (ELE) on sex-related vulnerability in developments of functional abdominal pain disorders (FAPDs) in 5-12 age group’’ Amaranath Karunanayake, Niranga Manjuri Devanarayana, Shaman Rajindrajith: The 14th Asian Pan-Pacific Society of Paediatric Gastroenterology, Hepatology and Nutrician Meeting, Bangkok, Thailand (2018) Abstract no. PP-G-16.

The conference proceeding was peer-reviewed and published as a poster /abstract presentation. Therefore, submitted work does not constitute dual publication Furthermore, the paper is not currently under consideration elsewhere for publication. We include a caption on this regard at the end of the manuscript. 

2. We make data are available within the supporting information files. Separate captions for supplementary files are included at the end of the manuscript.

---

## [Decision Letter · Decision Letter 3]

11 Jan 2022

PONE-D-20-15012R3Early life events in functional abdominal pain disorders in childrenPLOS ONE

Dear Dr. Karunanayake,

Thank you for submitting your manuscript to PLOS ONE. After careful consideration, we feel that it has merit but does not fully meet PLOS ONE’s publication criteria as it currently stands. Therefore, we invite you to submit a revised version of the manuscript that addresses the points raised during the review process. Specifically, the comments of Reviewer #3 still need to be addressed. 

We look forward to receiving your revised manuscript.

Kind regards,

Zhen Hua Hu, MD, PhD

Academic Editor

PLOS ONE

Reviewers' comments:

Reviewer's Responses to Questions

**Comments to the Author**

1. If the authors have adequately addressed your comments raised in a previous round of review and you feel that this manuscript is now acceptable for publication, you may indicate that here to bypass the “Comments to the Author” section, enter your conflict of interest statement in the “Confidential to Editor” section, and submit your "Accept" recommendation.

Reviewer #3: (No Response)

2. Is the manuscript technically sound, and do the data support the conclusions?

Reviewer #3: Partly

3. Has the statistical analysis been performed appropriately and rigorously? 

Reviewer #3: No

4. Have the authors made all data underlying the findings in their manuscript fully available?

Reviewer #3: Yes

5. Is the manuscript presented in an intelligible fashion and written in standard English?

Reviewer #3: Yes

6. Review Comments to the Author

Reviewer #3: The authors didn't address prior comments related to clarification and justification of Table 2 and used multivariate analysis. In the statistical analysis, authors stated "We used binary logistic regression to determine the independent association between early life events and FAPDs which were shown to be significant in the univariate analysis.". However, authors should define the cut-off for p value used to select the included items as well as adjust for other variables that showed significant statistical difference in the univariate analysis. In the multivariate analysis model, authors should report adjusted Odds Ratio (AOR), rather than just OR, in order to investigate the independent factors.

7. PLOS authors have the option to publish the peer review history of their article (what does this mean?). If published, this will include your full peer review and any attached files.

Reviewer #3: **Yes: **Elsayed Abdelkreem

---

## [Author Response · Author response to Decision Letter 3]

24 Mar 2022

Comments to the Author

1. If the authors have adequately addressed your comments raised in a previous round of review and you feel that this manuscript is now acceptable for publication, you may indicate that here to bypass the “Comments to the Author” section, enter your conflict of interest statement in the “Confidential to Editor” section, and submit your "Accept" recommendation. Authors response 

Reviewer #3: (No Response) 

2. Is the manuscript technically sound, and do the data support the conclusions?

Reviewer #3: Partly All the revisions proposed by the reviewers were addressed appropriately to generate a technically sound manuscript 

3. Has the statistical analysis been performed appropriately and rigorously? 

Reviewer #3: No The statistical analysis was improved in the revised manuscript which include the adjusted Odds Ratio (AOR) in the multivariate analysis model.

4. Have the authors made all data underlying the findings in their manuscript fully available?

Reviewer #3: Yes Thank you 

5. Is the manuscript presented in an intelligible fashion and written in standard English?

Reviewer #3: Yes Thank you 

6. Review Comments to the Author

Reviewer #3: The authors didn't address prior comments related to clarification and justification of Table 2 and used multivariate analysis. In the statistical analysis, authors stated "We used binary logistic regression to determine the independent association between early life events and FAPDs which were shown to be significant in the univariate analysis.". However, authors should define the cut-off for p value used to select the included items as well as adjust for other variables that showed significant statistical difference in the univariate analysis. In the multivariate analysis model, authors should report adjusted Odds Ratio (AOR), rather than just OR, in order to investigate the independent factors. The statistical analysis was improved in the revised manuscript which include the adjusted Odds Ratio (AOR) in the multivariate analysis model.

7. PLOS authors have the option to publish the peer review history of their article (what does this mean?). If published, this will include your full peer review and any attached files.

Do you want your identity to be public for this peer review? For information about this choice, including consent withdrawal, please see our Privacy Policy.

Reviewer #3: Yes: Elsayed Abdelkreem Your suggestions for improvements are highly appreciated

---

## [Decision Letter · Decision Letter 4]

19 Sep 2022

Early life events in functional abdominal pain disorders in children

PONE-D-20-15012R4

Dear Dr. Karunanayake,

We’re pleased to inform you that your manuscript has been judged scientifically suitable for publication and will be formally accepted for publication once it meets all outstanding technical requirements.

Kind regards,

Zhen Hua Hu, MD, PhD

Academic Editor

PLOS ONE

Additional Editor Comments (optional):

Reviewers' comments:

Reviewer's Responses to Questions

**Comments to the Author**

1. If the authors have adequately addressed your comments raised in a previous round of review and you feel that this manuscript is now acceptable for publication, you may indicate that here to bypass the “Comments to the Author” section, enter your conflict of interest statement in the “Confidential to Editor” section, and submit your "Accept" recommendation.

Reviewer #3: All comments have been addressed

2. Is the manuscript technically sound, and do the data support the conclusions?

Reviewer #3: Yes

3. Has the statistical analysis been performed appropriately and rigorously? 

Reviewer #3: Yes

4. Have the authors made all data underlying the findings in their manuscript fully available?

Reviewer #3: Yes

5. Is the manuscript presented in an intelligible fashion and written in standard English?

Reviewer #3: Yes

6. Review Comments to the Author

Reviewer #3: (No Response)

7. PLOS authors have the option to publish the peer review history of their article (what does this mean?). If published, this will include your full peer review and any attached files.

Reviewer #3: **Yes: **Elsayed Abdelkreem

---

## [Editor Report · Acceptance letter]

24 Oct 2022

PONE-D-20-15012R4 

Early life events in functional abdominal pain disorders in children 

Dear Dr. Karunanayake:

I'm pleased to inform you that your manuscript has been deemed suitable for publication in PLOS ONE. Congratulations! Your manuscript is now with our production department. 

Kind regards, 

on behalf of

Dr. Zhen Hua Hu 

Academic Editor

PLOS ONE